# Fibonacci spirals may not need the Golden Angle

Xiaofeng Yin[1,2]* and Hirokazu Tsukaya[1]

[1]Department of Biological Sciences, Graduate School of Science, The University of Tokyo, Tokyo, Japan; [2]Japan Society for the Promotion of Science, Tokyo, Japan

## Insights

auxin; capitulum; parastichy; pattern transition; phyllotaxis.

**Author for correspondence:**
X. Yin,
E-mail: yinx@g.ecc.u-tokyo.ac.jp

### Abstract

Phyllotaxis, the regular arrangement of plant lateral organs, is an important aspect of quantitative plant biology. Some models relying on the geometric relationship of the shoot apex and organ primordia focus mainly on spiral phyllotaxis, a common phyllotaxis mode. While these models often predict the dependency of Fibonacci spirals on the Golden Angle, other models do not emphasise such a relation. Phyllotactic patterning in Asteraceae is one such example. Recently, it was revealed that auxin dynamics and the expansion and contraction of the active ring of the capitulum (head) are the key processes to guide Fibonacci spirals in gerbera (*Gerbera hybrida*). In this *Insights* paper, we discuss the importance of auxin dynamics, distinct phases of phyllotactic patterning, and the transition of phyllotaxis modes. These findings signify the local interaction among primordia in phyllotactic patterning and the notion that Fibonacci spirals may not need the Golden Angle.

The regular arrangement of plant lateral organs, known as phyllotaxis, has been the subject of quantitative plant biology research for a long time (Adler et al., 1997; Barabé & Lacroix, 2020; Jean & Barabé, 1998; Yin, 2021). Yet, there remains many unresolved questions (Yin & Kitazawa, 2021), such as the rapid and *de novo* phyllotactic patterning in the capitulum (head) of Asteraceae. Spiral and whorled are the two major phyllotaxis modes (Yin, 2021). In the spiral mode, two families of eye-catching winding spirals called parastichies (Figure 1a) can be observed (Jean, 1994). The numbers of parastichies are often successive numbers of Fibonacci series <1, 1, 2, 3, 5, 8, 13, 21, 34...> (Figure 1a). Neighbouring organs along a parastichy, nonetheless, are not 'neighbours' in terms of ontogeny. The next organ is formed in a different parastichy and usually forms a divergence angle (Figure 1a) with the previous one by 137.51°, which is the golden fraction of 360° and thus called the Golden Angle.

More than 150 years ago, Wilhelm Hofmeister made the pioneering postulation that a new organ is formed at the farthest position from existing ones on the growing shoot apex (Hofmeister, 1868). Mary and Robert Snow tested and elaborated this postulation and suggested that a new organ is formed as soon as a minimum space becomes available (Snow & Snow, 1931; 1933; 1935; 1947). This repeating process can be conceptualised as a self-organising process of stacking new elements to existing ones on a circle or a cylinder (Atela, 2011; Atela et al., 2003; Douady & Couder, 1996a; 1996b; 1996c; Douady & Golé, 2016; Godin et al., 2020; Golé et al., 2016; Mitchison, 1977). Various models suggest that geometric constraints of the shoot apex and organ primordia canalise phyllotaxis towards Fibonacci spirals (Battjes et al., 1993; Godin et al., 2020). In a special case, when assuming the Golden Angle as the divergence angle, Fibonacci spirals are the only solution, mathematically (Battjes & Prusinkiewicz, 1998; Battjes et al., 1993; Fowler et al., 1989; 1992; Godin et al., 2020; Hirmer, 1931).

Is the Golden Angle necessary for Fibonacci spirals to form? Not necessarily. Various models do not make *a priori* assumptions related to the Golden Angle, yet they can generate Fibonacci spirals (Atela, 2011; Atela et al., 2003; Douady & Couder, 1996a; 1996b; 1996c; Golé et al., 2016; Mitchison, 1977). Notably, combining empirical and theoretical efforts, a recent study further revealed that the *de novo* establishment of Fibonacci spirals in the capitulum of gerbera (*Gerbera hybrida*) does not require the Golden Angle (Zhang et al., 2021). A gerbera capitulum consists of outer bracts and inner florets on a receptacle (Figure 1a; Zhang et al., 2021). At the beginning of

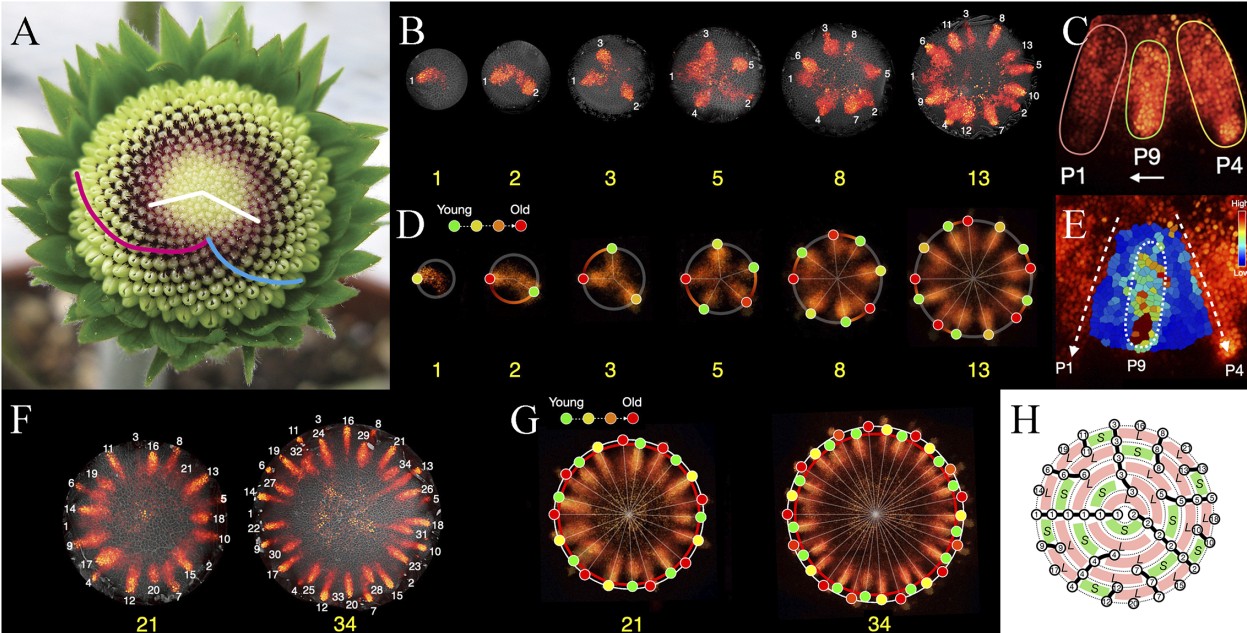

**Fig. 1.** Key patterning processes in gerbera (*Gerbera hybrida*) capitula development. (a) A capitulum (head) of *G. hybrida* with outer bracts and inner florets. Pink spiral, one out of 21 clockwise parastichies. Blue spiral, one out of 34 counter-clockwise parastichies. Note that 21 and 34 are two successive numbers of Fibonacci series. White lines, divergence angle of two florets. (b) Confocal images of DR5 reporter indicating that, at the beginning of the patterning process, auxin maxima emerge at approximately the same radial distance from a few discrete steps. (c) Details of DR5 patterning of a newly emerged maximum. The new maximum (P9) moves laterally towards its older neighbour (P1). (d) Predicted distribution of bract primordia (coloured dots) overlaid on confocal images of DR5 reporter. (e) Quantification of DR5 signal intensity in the same region as in (c). (f) Confocal images of DR5 reporter indicating that, as the patterning process continues, auxin maxima emerge closer to the centre. (g) Zigzag-like pattern front formed by initially emerged auxin maxima (on the white circle) and later emerged auxin maxima (on the red circle). Note the slight size difference between the white and the red circle. (h) A schematic diagram showing the lateral movement of auxin maxima. As a result, a long (*L*) gap and a short (*S*) gap are generated. Note that the lateral movement is always towards the older neighbour. White numbers in (b,f) and numbers in (h) indicate only positions and do not imply the order of emergence; yellow numbers in (b,d,f,g) indicate the total number of auxin maxima. (a) Modified from Elomaa (2017). (b–g) Modified from Zhang et al. (2021). (h) Derived from Zhang (2021).

the patterning process, up to 13 bract primordia emerge at the capitulum rim in an approximately circular pattern with negligible radial distance differences, as indicated by DR5 promoter (an auxin signalling output reporter) (Figure 1b; Zhang et al., 2021). Curiously, these auxin maxima are not formed altogether at once but rather formed from a few discrete steps (Figure 1b; Zhang et al., 2021). In each step, the number of added maxima is the next Fibonacci number, which then quickly leads the number of total maxima to become Fibonacci numbers (Figure 1b; Zhang et al., 2021). Time-lapse imaging reveals one unique feature of auxin maxima dynamics that each maximum emerges at approximately halfway in between existing ones and subsequently moves laterally (Figure 1c,e,h; see Zhang et al., 2021, fig. 5). Most importantly, the lateral movement is always towards its older neighbour and the total distance travelled could be as many as six to eight cells (Figure 1c,e,h; see Zhang et al., 2021, fig. 5). The importance of lateral movement of auxin maxima is further explored by a computational model approximated as a growing circle. Without lateral movement, the circular model predicts that, in each step, the number of auxin maxima formed would be 1, 2, 4, 8, … instead and these maxima would be evenly distributed around the rim (see Zhang et al., 2021, fig. S5). Such evenly distributed maxima could be considered as whorled modes (Yin, 2021). With a certain range of lateral movement values towards the older neighbour, in contrast, the circular model predicts that the number of auxin maxima tends to be a Fibonacci number and maxima positions faithfully match the experimental observations (Figure 1d; Zhang et al., 2021). Thus, without any *a priori* assumptions related to the Golden Angle, the initial establishment of Fibonacci spirals can be robustly achieved.

As the patterning process continues, aside from the lateral movement, new auxin maxima are now positioned slightly closer to the centre of the capitulum (Figure 1f; Zhang et al., 2021). This is because the expansion rate of the competent zone for primordia initiation, the active ring, cannot keep pace with the growing capitulum (see Zhang et al., 2021, fig. 6). Consequently, the active ring dissociates from the rim and the new auxin maxima form a zigzag-like pattern front (Figure 1g; Zhang et al., 2021). Following the gradual contraction of the active ring, the zigzag patterned auxin maxima are further propagated into a lattice on the capitulum. Parastichy numbers of this lattice decrease following the reversed order of Fibonacci sequence from outside to inside (Zhang et al., 2021). Eventually, the pattern becomes chaotic in the centre due to limited space as the entire capitulum is consumed (Zhang et al., 2021). Combining features including lateral movement of auxin maxima, differential dynamics between the active ring and the capitulum, and the actual shape of the receptacle, a three-dimensional integrative model can robustly simulate the entire patterning process (Zhang et al., 2021). Strikingly, Fibonacci spirals can still be generated even when the active ring loses its radial symmetry, which is commonly observed in irregular and fasciated gerbera heads (Prusinkiewicz et al., 2022; Zhang et al., 2021). This indicates that phyllotactic patterning does not rely on the overall symmetry of the active ring but rather on the local interactions among primordia (Golé et al., 2016; Zhang et al., 2021). 'Divergence section', the arc length of the active ring covered by two successive primordia, is the potentially significant parameter in asymmetric systems (Prusinkiewicz et al., 2022).

Many models of phyllotaxis assume that the positions of auxin maxima are fixed once they are formed (Bayer et al., 2009; Jonsson et al., 2006; Smith et al., 2006a). Following this assumption, phyllotaxis research mainly focuses on mechanisms that determine the positions of auxin maxima. Many factors including auxin concentration (Reinhardt et al., 2000; 2003), auxin signalling (Bhatia et al., 2016), auxin flux (Abley et al., 2016; Stoma et al., 2008), auxin biosynthesis (Galvan-Ampudia et al., 2020; Pinon et al., 2013; Yoshikawa et al., 2014), cytokinin signalling (Lee et al., 2009), mechanical signals (Feraru et al., 2011; Hamant et al., 2008; Heisler et al., 2010), and cell wall signatures (Braybrook & Peaucelle, 2013) are found to play a role in instructing the polarity of PIN1 proteins, which in turn guide the direction of auxin flow and the positions of auxin maxima (Heisler et al., 2005; Okada et al., 1991; Reinhardt et al., 2000; 2003). More recently, it was revealed that local PIN1 polarity converges towards the radial auxin maxima movement route in Arabidopsis inflorescence meristem and the duration of cell exposure to auxin is another important factor in phyllotaxis (Galvan-Ampudia et al., 2020). In gerbera capitula, the discovery of lateral auxin maxima movement before any physical appearance of primordia initiation adds another layer of complexity concerning primordia's 'prehistory'. Backed up by computational models, it was shown that such 'prehistory' is the contributing factor that leads to the initial asymmetry and Fibonacci numbers of parastichies (Zhang et al., 2021). Are there any additional factors that instruct PIN1 polarity in gerbera? How does PIN1 facilitate the lateral auxin maxima movement? And most importantly, why do the lateral auxin maxima always move towards the older neighbour and what is the cue to break the initial symmetry? Answers to these questions would further enrich our understanding of phyllotactic patterning in such a system.

Another implication of lateral auxin maxima movement is that it may prompt the transitions of phyllotaxis modes. The circular model predicts that without the lateral movement, whorled modes would appear instead of the spiral mode (Zhang et al., 2021). Most eudicots experience at least one transition from whorled to spiral during their ontogeny (Meicenheimer, 1998). The transition between phyllotaxis modes can be explained as a non-uniform change of the apex (Meicenheimer, 1998; Meicenheimer & Zagórska-Marek, 1989); changes of a suite of factors including auxin concentration and biosynthesis (Smith et al., 2006a); and an age-dependent decrease of primordia inhibition field (Smith et al., 2006b). Lateral auxin maxima movement hence offers another possible explanation for the transitions between phyllotaxis modes. It is also of interest to note that distances of bracts to the stem centre are clustered upon maturity and each cluster corresponds to a discrete step of auxin maxima burst at the beginning of the patterning process (Zhang et al., 2021). This interesting pattern of bract primordia and their corresponding auxin maxima has properties of both whorled and spiral modes. On the one hand, the radial positioning of auxin maxima follows the whorled mode such that each step, or each individual auxin maxima burst, could be considered as a distinct 'whorl'. On the other hand, the whorled mode has evenly distributed organ whereas the angular spacing of auxin maxima is not even and clearly following the spiral mode. This suggests that it might be an intermediate or transitional state between the spiral mode and the whorled mode. What, then, could cause this? One hint comes from a phenomenon known as 'permutation', an incorrect order of organs at correct angular positions. Permutation is found in sunflower (Couder, 1998) as well as various Arabidopsis accessions and mutants (Besnard et al., 2014; Fal et al., 2017; Landrein et al., 2015). Permutation

results from locally disrupted timing of organ initiation such as co-initiation while organ angular positions remain unaffected (Besnard et al., 2014). Especially, when the size of primordia is small enough as compared to the apex, which is an attribute of gerbera capitula, the frequency of permutation would be high (Landrein et al., 2015; Mirabet et al., 2012). Nonetheless, the most common permutations involve a series of two or three organs in Arabidopsis inflorescence meristem (Besnard et al., 2014), whereas as many as eight simultaneous auxin maxima could form in gerbera capitula at the beginning of the patterning process. Regardless, it is extremely intriguing to further dissect the mechanisms that lead to this 'hybrid' pattern to deepen our understanding on the transition between phyllotaxis modes.

Phyllotaxis as a self-organising process of stacking new elements is well characterised down to the molecular level; and models with this feature often emphasise the relationship between the Golden Angle and Fibonacci spirals. In sharp contrast to this paradigm, various models do not make *a priori* assumptions related to the Golden Angle, yet they can generate Fibonacci spirals. Phyllotactic pattering in gerbera capitula offers yet another compelling example. After all, Fibonacci spirals may not need the Golden Angle. This urges us to reconsider phyllotaxis as a simple patterning process depending on local interactions.

## Acknowledgements

We thank Paula Elomaa for critical reading on an earlier version of the manuscript; Paula Elomaa and Teng Zhang for sharing figures and intriguing discussion on unpublished results and two reviewers for their insightful suggestions.

**Financial support.** This work is supported by Ministry of Education, Culture, Sports, Science and Technology (MEXT), Japan (Scientific Research on Priority Areas and Scientific Research on Innovative Areas, Grant No. JP 19H05672) and Japan Society for the Promotion of Science Postdoctoral Fellowship for Research in Japan (Grant No. P19085).

**Conflict of interest.** The authors declare no conflict of interest.

**Authorship contributions.** X.Y. conceived the paper. X.Y. and H.T. wrote the paper.

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
