## [Reviewer Report]

Dear Prof. Hamant,

We wish to submit our manuscript, entitled “Fibonacci spirals may not need the Golden Angle”, to *Quantitative Plant Biology*, as an *Insights* paper.

Phyllotaxis, the arrangement of plant lateral organs, is a long-standing topic in the field of quantitative plant biology. It has been assumed that the appearance of Fibonacci spirals depends on the Golden Angle between successive organs in a global scale. However, using approaches such as quantitative live imaging and computer modeling, it was recently revealed that local, instead of global, interactions among primordia play the key role in phyllotactic patterning in the capitula (heads) of Asteraceae (Zhang et al., 2021. *PNAS*). In our manuscript, we give an overview of this work, highlight the importance of lateral auxin maxima movement, and discuss its potential implications in phyllotactic transitions. We believe that our manuscript is timely and well fit the scope of *Quantitative Plant Biology*.

Best wishes,

Xiaofeng Yin, Ph.D.

Department of Biological Sciences

Graduate School of Science

The University of Tokyo

Hongo 7-3-1, Bunkyo-ku, Tokyo, Japan. 113-0033

---

## [Reviewer Report]

*Comments to Author*: In this manuscript from Xiaofeng Yin and Hirokazu Tsukaya, the authors propose a new mechanism behind phyllotaxis formation involving the movement of newly formed auxin maxima dispensing with mathematical concept like the golden angle. The authors show data extracted from the model organism Gerbera hybrida comprising auxin quantification obtained from the DR5 reporter which helped them to identify auxin maxima around the apex and also performed a computer simulation of their proposed model.

I have some questions and observations regarding the evidence shown in the paper and its consequences:

- In Fig. 1C, the authors claim the presence of movement of maximum P9 towards its older maximum P1. However, to appreciate the displacement, it would be necessary to show P9 before its movement, maybe after its formation, or a time lapse of the same.

- Is it possible to quantify this auxin maxima movement phenomenon? Maybe in terms of distance travelled along the apex? When does a maximum move, immediately after formed? Is it an abrupt move or slow transition?

- Lines 74-76: I think it is necessary show this simulation of the model without auxin movement. Also, I am not sure about how are materials and methods presented in this kind of "Insights" paper, but model details (how the model has been made) should be presented.

- Lines 91-94: If the authors are suggesting the possible future development of a computer model, they should replace "a three-dimensional integrative model CAN robustly" with "a three-dimensional integrative model COULD robustly" as we cannot possible predict the outcome of an experiment beforehand.

- Lines 94-97: I am not an expert in gerbera, but maybe some data regarding the dynamics of the active ring should be presented and compared with the auxin data, as they claim that its symmetry is not necessary for the phyllotaxis formation.

---

## [Reviewer Report]

*Comments to Author*: This is a nice synopsis of experimental/modeling work about the inception of Fibonacci patterns, especially in compositae inflorescences, up to the PNAS paper of Zhang et. al. (2021). This referee agrees with the fact that evidence in this paper shows that Fibonacci phyllotaxis does not need the Golden Angle a priori and that these pattern emerge from local interactions between primordia. But the referee is not sure that Zhang et. al. (2021) represents a departure in proving this thesis, which seems to be one of the central tenets of the article being refereed here.

Indeed, this referee differs with the authors in what they seem to imply: that only now, after the paper of Zhang et. al.(2021) do we understand why models do not need to assume the Golden Angle for the inception of Fibonacci patterns. In fact, most of the modeling work cited by the authors (Atela, 2011; Atela et al., 2003; Douady & Couder, 1996a, b, c; Douady & Golé, 2016; Godin et al., 2020; Golé et al., 2016; Mitchison, 1977) do not assume a priori a divergence angle close to the Golden Angle to generate Fibonacci patterns, and while they do often observe a convergence to that angle after variation of the growth parameter (ratio primordia size/meristem size), they also observe that, many times, this is not the case, even if the resulting pattern is Fibonacci. So, whereas it is important to point out, as the authors do, the all-too common misconception that the Golden Angle is the organizing principle behind Fibonacci phyllotaxis, this fact has been recognized and observed many times before, and for at least 25 years.

The observation of lateral motion of initia is something that seems new with Zhang et. al.’s work (albeit one needs to worry about the fact that the motion of these initia is detected via markers of auxin, not auxin itself). But this is not altogether in contrast with previous models. It could be seen as adding an interesting piece of prehistory to the positioning of primordia, rather than contradict the fact that, overall, primordia stay radially put once they have really formed, and that these positions determine in fine the position of the new primordium.

In other words, whereas this referee is ready to support the idea that the rich and interesting article of Zhang et. al. (2021) confirms what was known – that in plants, Fibonacci does not need thee Golden Angle and that phyllotaxis is a self-organizing process guided by local biochemical interactions, the referee is not sure that Zhang et. al. offers a “sharp contrast of paradigm” in that respect.

Since this sharp contrast seems to be a central part of this insight, this referee would like the messaging to be re-worked before this article is accepted.

In the meantime, here are some suggested corrections for minor issues in the paper.

39 Parastichies do not necessarily meet near/at 90 deg angles, even at their inception.

43 137.51 deg. (not 137.49)

59 Ambiguous phrase, that seems to contradict the main point of the authors: “Can Fibonacci spirals form without the Golden Angle? Not necessarily.” Consider an option such as: “Is the Golden Angle necessary for Fibonacci spirals to form? Not necessarily.”

80 Thus, without any prior assumptions related to the Golden Angle, the initial establishment of Fibonacci spirals can be robustly achieved.

121 why does the lateral auxin

129 “unequal change of the apex”. Do the authors mean “non-uniform” or “non-isotropic”?

147 Review the confusing grammar of “Especially, when the primordia …”

---

## [Reviewer Report]

*Comments to Author*: The authors need to be careful in interpreting the literature. For example, many existing models, including the ones cited in this study, do not necessarily assume a priori a divergence angle. I would also like to suggest the authors to reconsider the interpretation of the observed lateral motion of DR5 signals (Zhang et al., 2021 PNAS and Galvan-Ampudia et al., 2020 eLife).

---

## [Reviewer Report]

*Comments to Author*: This is an improvement over the first version.

A couple of small detail:

This referee does not believe Godin et. al. (2020) predicates a canalization towards the Golden Angle, but rather to Fibonacci phyllotaxis - two independent things as this paper is trying to establish. Also, spelling line 59: Various models do not make a priori assumptions relateD to the Golden Angle